# An optical reaction micro-turbine

Silvio Bianchi[1], Gaszton Vizsnyiczai[2,3], Stefano Ferretti[2], Claudio Maggi[1] & Roberto Di Leonardo[1,2]

To any energy flow there is an associated flow of momentum, so that recoil forces arise every time an object absorbs or deflects incoming energy. This same principle governs the operation of macroscopic turbines as well as that of microscopic turbines that use light as the working fluid. However, a controlled and precise redistribution of optical energy is not easy to achieve at the micron scale resulting in a low efficiency of power to torque conversion. Here we use direct laser writing to fabricate 3D light guiding structures, shaped as a garden sprinkler, that can precisely reroute input optical power into multiple output channels. The shape parameters are derived from a detailed theoretical analysis of losses in curved microfibers. These optical reaction micro-turbines can maximally exploit light's momentum to generate a strong, uniform and controllable torque.

[1] NANOTEC-CNR, Institute of Nanotechnology, Soft and Living Matter Laboratory, Roma I-00185, Italy. [2] Dipartimento di Fisica, Università di Roma "Sapienza", Roma I-00185, Italy. [3] Institute of Biophysics, Biological Research Centre Hungarian Academy of Sciences, Szeged H-6726, Hungary. Correspondence and requests for materials should be addressed to R.L. (email: roberto.dileonardo@uniroma1.it)

Turbines are a large class of turbomachines that transfer energy from a working fluid, typically water, air, or steam, to a continuously rotating system, the rotor. Despite the large variability in turbine designs and working fluids, torque generation always results from the continuous angular momentum change of the working fluid that flows through them. In the same way, microscopic turbines can use light as a working fluid and generate a torque by redirecting the flow of optical energy through them so that outgoing light has a different angular momentum than incoming light. Both for a macroscopic hydraulic turbine or a microscopic optical turbine the generated torque will be then given by the difference in the rate of incoming and outgoing angular momentum, that in the optical case reads:

$$T_z = \frac{Pn}{c}\left(k_\phi^{\text{out}} r^{\text{out}} - k_\phi^{\text{in}} r^{\text{in}}\right) \qquad (1)$$

where $P$ is the optical power, $n$ is the refractive index of the surrounding medium, $z$ is the direction of the rotor axis, $\phi$ the azimuthal coordinate, $k_\phi^{\text{in}}$ and $k_\phi^{\text{out}}$ the azimuthal components of respectively incoming and outgoing light directions and $r^{\text{in}}$ and $r^{\text{out}}$ the radial distances of inlet and outlet (Fig. 1a). Equation (1) is formally very similar to the Euler turbomachine equation[1] expressing torque in hydraulic turbines and where $P$ is replaced by mass flow and $nk_\phi^{\text{in}}/c$, $nk_\phi^{\text{out}}/c$ by the azimuthal components of inlet and outlet velocities. In the optical case, however, although incoming light will often be a focused laser beam with a well defined inlet point and propagation direction, outgoing light results from a complex scattering interaction and cannot be associated with unique values of $k_\phi^{\text{out}}$ and $r^{\text{out}}$. However, because of its fundamental character, Eq. (1) can still be used to put an upper bound to the torque that an optical turbine can generate. In the case of axial turbines where incoming light propagates in the $z$ direction ($k_\phi^{\text{in}} = 0$), the maximum achievable torque is:

$$T_{\text{max}} = \frac{Pnr}{c} \qquad (2)$$

corresponding to all outgoing light emerging with tangential direction ($k_\phi^{\text{out}} = 1$) from the turbine point having the largest distance from the axis ($r$). Different designs for optical turbines have been proposed in recent literature. The so-called "light mill"[2,3] is an axial turbine with a propeller-shaped 3D micro-rotor. A central rod component guarantees stable trapping and alignment along the axis of a focused tweezer beam while tilted radial arms scatter light with a nonzero component of orbital angular momentum. In ref. [2] it is shown that a beam with power $P = 10$ mW could spin a $r = 3\,\mu$m structure at an angular frequency of $\omega = 7$ rad s$^{-1}$. The corresponding optical torque will balance the hydrodynamic viscous torque $T = \gamma\omega \sim 1$ pN μm where we used for the rotational drag coefficient $\gamma = 0.2$ pN μm s rad$^{-1}$ as estimated for a very similar structure in ref. [2]. Using Eq. (2) we find that the maximum achievable torque is $T_{\text{max}} = 136$ pN μm. The resulting torque efficiency, defined as the ratio $T/T_{\text{max}}$ is <1%. Similarly to light-mills, micro-structures of different shapes can exploit chirality to generate torque. Among them, cylinders[4] with slanted bases or chiral crosses[5,6] could only generate a torque that is two orders of magnitude smaller than the maximum value.

Another proposed design for an optical turbine is the paddle-wheel[7] that, as an optical equivalent of a Pelton turbine, employs a non-chiral structure and an off-axis pushing beam. The structure consists of four paddles connected to a central axis with two spheroids at its extremities. The two spheroids serve as handles for optical trapping while the paddles recoil by partially reflecting a low divergence beam thus causing the rotation of the entire structure. From the reported values for drag

$\gamma = 5.3$ pN μm s rad$^{-1}$, power $P = 6$ mW and angular speed $\omega = 1.8$ rad s$^{-17,}$] we estimate an optical torque of $T = \gamma\omega = 9.5$ pN μm. This design results in an improved torque efficiency (10% of the maximal value $T_{\text{max}} = 106$ pN μm, excluding the power in the trapping beams) although rotations are strongly non uniform. Other optical turbines, employing symmetric rotors like the paddle-wheel, rely on structured wavefront of the beam to generate torque. Laguerre–Gauss beams have been employed for this purpose giving a ratio between torque and $T_{\text{max}}$ that is comparable[8] (~0.01) or lower[9] (~$10^{-3}$) than the above cited cases.

Finally, birefringent particles can generate torque through the exchange of the spin component of optical angular momentum[10–13]. The maximum achievable torque in this case is $P\lambda/\pi c$ that is only a fraction $\lambda/\pi nr$ of the maximum torque in Eq. (2) (from ref. [13] we compute a torque efficiency of the order of $10^{-3}$). Despite a small torque efficiency, high rotation rates can be still obtained with this method provided that the rotational drag is very low as in the case of small particles[11] or low viscosity fluids.[12,13] Here we are interested in maximizing the efficiency of conversion from optical power to torque so that, in the following, we will only focus on strategies exploiting the orbital angular momentum of light.

Both the light-mill and the paddle-wheel are close analogues of hydraulic impulse turbines where the working fluid, moving in free-space, impacts on the turbine blades transferring impulse. In a different class of turbines, called reaction turbines, the working fluid is always enclosed in the blade system which, by deflecting the incoming mass flow, recoils by reaction forces.

Here we designed a light-guiding structure made of SU8 photoresist with a chiral shape resembling that of a garden sprinkler. By replacing scattering with guided light propagation inside micro-fibers we could reroute most of the incoming light onto outlet points located at the radial periphery of the structure and from which light emerges in a tangential direction. With such a design we maximize the reaction torque that is in the same order of magnitude of the ideal maximum value.

## Results

**Bend losses.** The structure starts with a larger circular core (4 μm) oriented in the axial $\hat{z}$ direction (see Fig. 1c). The input fiber then splits into four 2 μm cores that guide light towards outlet fiber tips oriented in the azimuthal direction $\hat{\phi}$ (Fig. 1c). If all available light could be coupled into the structure and propagate all the way to the outlets with no losses, this design would achieve the theoretical maximal torque in equation (2). We know, however, that three main causes of losses are present in our structures: coupling, splitting, and bending.

Bend losses, although always present in curved optical fibers, decrease exponentially with the curvature radius $R$[14,15] becoming negligible for macroscopic bends ($R \sim 1$ cm). However, a microscopic optical turbine will need to redirect incoming light within a small propagation distance by using optical fibers with curvature radii in the micron range. Although bend losses have been widely studied in literature[15–17], scaling those results to the range of micrometric curvature radii is not straightforward and potentially inaccurate due to breaking of typical approximations.

To get a direct and accurate evaluation of bend losses, we used direct laser writing via two-photon polymerization[18,19] to fabricate a set of 180° arches having a constant radius of curvature $R$ and a circular cross-section with diameter $2a = 2\,\mu$m. Both the input and output facets of the fiber arches lie on the coverslip surface plane allowing us to use the same objective (20 × NA = 0.5) to couple light in and collect output light. A laser beam ($\lambda = 1064$ nm) is focused at the center of the input core while the transmitted light is imaged with a digital camera (Fig. 2a).

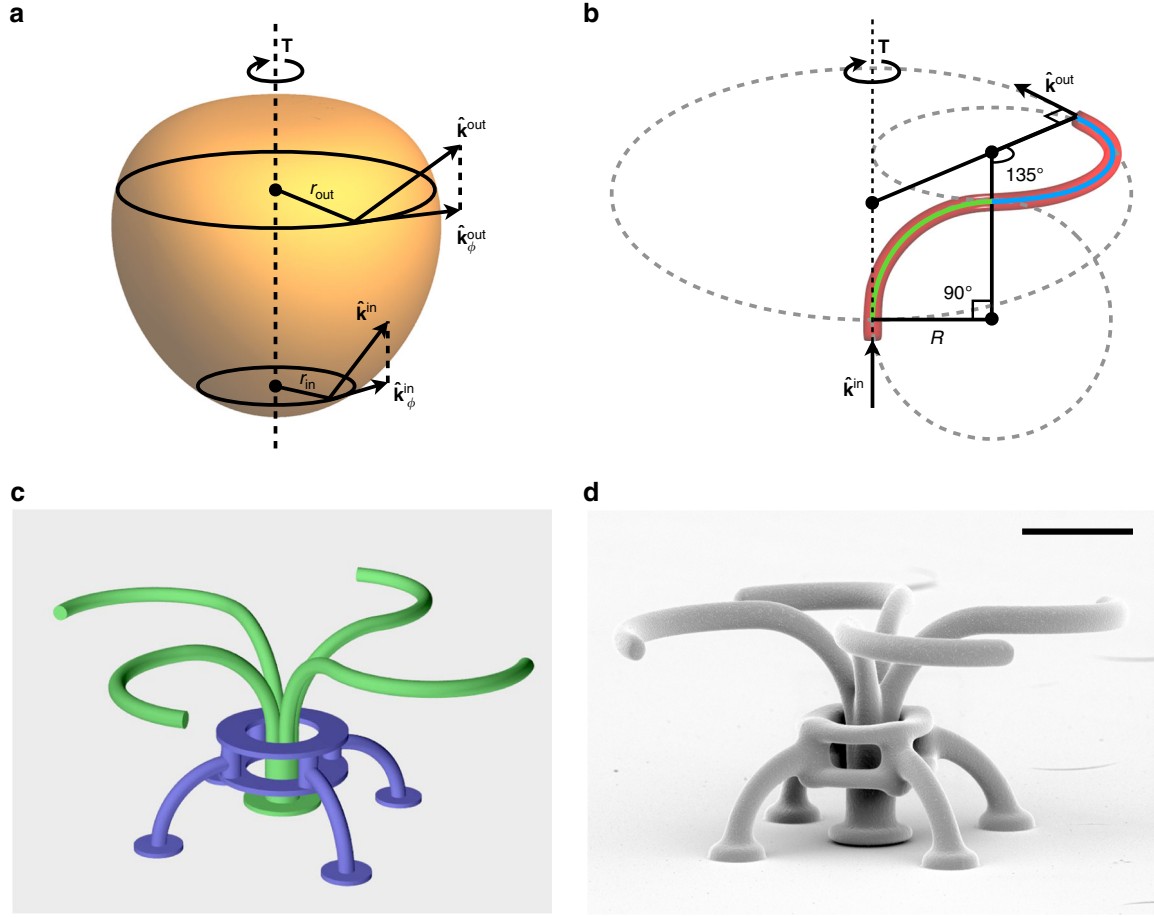

**Fig. 1** Microturbine design. **a** A schematic representation of the generic working principle of torque generation in turbines. The working fluid enters and leaves the rotor with directions and radial distances indicated respectively with $\hat{\mathbf{k}}^{in,out}$ and $r^{in,out}$. The optical torque is given by the difference between the input and output flux of angular momentum $T = (nP/c)(\hat{\mathbf{k}}^{out} \times \mathbf{r}^{out} - \hat{\mathbf{k}}^{in} \times \mathbf{r}^{in})$. **b** 3D shape of the curved arms. The arm is composed of two circular arcs: the first (green) redirects light from axial to radial direction while the second (blue) from radial to tangential. **c** 3D computer model of the designed structure. The stator, which is anchored to the cover glass, is highlighted in blue while the rotor, composed by a waveguide splitting into four curved arms, is highlighted in green. **d** Scanning electron microscopy image of a micro-turbine (scale bar is 10 μm)

Transmitted power is recorded by integrating light intensity over a circular region that is about twice the size of the fiber core. We repeated the procedure for 5 arches having the same curvature radius and for a total of 10 radii. The laser power that is actually coupled to each fiber arc is unknown but the same for every $R$ since it only depends on the core diameter and input beam shape. Therefore we only know transmission versus $R$ within a common normalization factor that is chosen to best match theory and simulation at large $R$ values (see Fig. 2b). A strong decrease in transmission is observed for curvature radii smaller than 10 μm while a slower decrease is observed at larger radii. A similar two step decay is obtained using the analytical formula derived in ref. [15], (dashed line in Fig. 2b) a refined version of the theory proposed by Marcuse[14]. The formula predicts losses in each mode so we first need to find excited modes in our fibers. If straight, our fibers would support three guided modes ($LP_{01}$, $LP_{02}$, and $LP_{11}$) of which only the first two are excited by our focused laser beam (NA = 0.5, FWHM = 2.3 μm), with the first mode taking about 70% of guided light. The largest drop of transmission at about $R = 10$ μm is due to the first mode ($LP_{01}$) while the slow transmission decrease at larger $R$ is due to the second mode ($LP_{02}$).

When comparing quantitative values, however, strong deviations between theory and experiment are found for $R < 20$ μm. This is not so surprising when we consider that analytical predictions were based on approximations like weak guiding and large curvature radius ($R \gg a$) which may not be valid in the present case. A closer theoretical description can be obtained by numerical simulations with 3D scalar Beam Propagation Method (BPM)[20]. We first write the Helmholtz equation for the field $U$ using a cylindrical coordinate system ($\rho$, $\phi$, $z$) as shown in Fig. 2a:

$$\left[\frac{\partial^2}{\partial z^2} + \frac{1}{\rho^2}\frac{\partial^2}{\partial \varphi^2} + \frac{1}{\rho}\frac{\partial}{\partial \rho}\rho\frac{\partial}{\partial \rho} + n^2 k_0^2\right] U = 0 \quad (3)$$

where $k_0 = 2\pi/\lambda$ is the wavenumber in vacuum. A common method for simulating a bent waveguide with BPM prescribes a coordinate transformation from the cylindrical coordinate system to a curvilinear coordinate system ($u = R\log(\rho/R)$, $s = R\phi$, $z' = z$). Under such a transformation, the Helmholtz equation is written as:

$$\left[e^{2u/R}\frac{\partial^2}{\partial z^2} + \frac{\partial^2}{\partial s^2} + \frac{\partial^2}{\partial u^2} + \left(e^{u/R}n\right)^2 k_0^2\right] U = 0 \quad (4)$$

The coordinate transformation has the effect of mapping $n$ onto an effective refractive index $ne^{u/R}$ (see Fig. 2a). As the effective refractive index increases with $u$, evanescent tails of the guided modes can leak outside of the core leading to

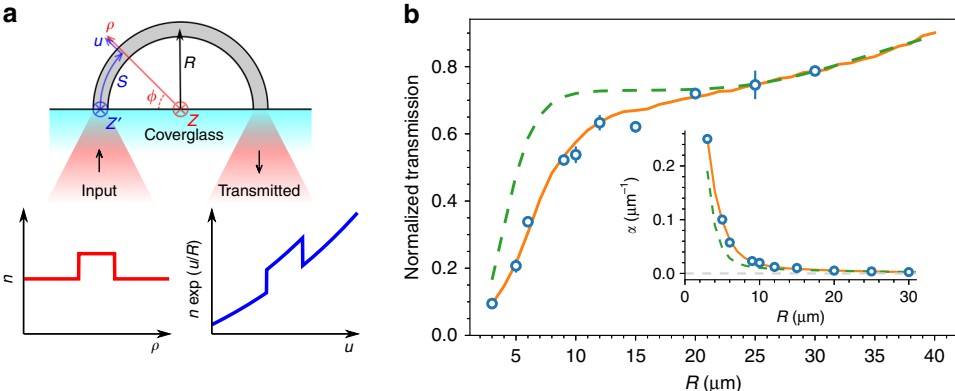

**Fig. 2** Bend losses. **a** An infrared laser beam is coupled into one end of a 180° fiber arc to measure attenuation of light transmitted to other fiber end. The red line represents the refractive index ($n$) profile along the radial coordinate $\rho$. Propagation in curved fibers can be mapped into a straight fiber problem with the effective refractive index shown by the blue line. **b** Transmitted power as a function of the arc curvature radius. Experimental data are shown as circles. Error bars represents the standard deviation between the transmission of 5 arches with same curvature. BPM simulations and analytical prediction are plotted as solid orange and dashed green lines respectively. Inset shows the attenuation coefficient per unit length obtained from transmission data in the main panel. Gray dashed line is a guide to the eye representing the axis $\alpha = 0$

power losses. If the term multiplying the partial derivatives in $z$ is dcneglected (i.e., $e^{2u/R} \frac{\partial^2}{\partial z^2} \approx \frac{\partial^2}{\partial z^2}$), the system can be simulated using a standard BPM algorithm[20] where the propagation direction is $s$ while $u$ and $z'$ are discretized on a grid. However, since our waveguides have a large curvature (small $R$), we modified the BPM algorithm to simulate Eq. (4) without approximations. Results are plotted as solid lines in Fig. 2b showing a remarkably good agreement with experimental findings.

**Design of an optical turbine**. To design an efficient optical turbine, we need solid microfibers tracing 3D paths that maximize reaction torque while minimizing bend losses. The rotor was designed as a four-arm structure where each arm is composed by two arcs with constant curvature radius $R$. The first one, appearing in green in Fig. 1b, spans 90° and diverts axially incoming light in the radial direction. The second arc, in blue, lies on a plane that is orthogonal to the axial direction and extends for 135° ($3\pi/4$) turning light's propagation direction from radial to azimuthal. As a result the radial distance of the fiber outputs from the structure axis is $r = (1 + \sqrt{2})R$. The total light emerging from the four fiber outputs will be $P = \beta\sigma\chi P_0$ with $P_0$ the input power and $\beta$, $\sigma$, $\chi$, respectively, the attenuation due to bending, splitting, and coupling losses. For a fixed number of arms and fiber core diameters $\sigma$ and $\chi$ will be constant while $\beta(r)$ will be a function of system size. Analysis of bend losses in our fibers suggests that curvature radius must be kept above 10 μm to avoid severe bend losses. However, a much larger structure size would result in slower rotations and a smaller mechanical power output due to increasing viscous drag. To see this we recall from (1) that while the applied torque will scale as $T \sim P(r)r \sim \beta(r)r$ the rotational drag $\gamma$ scales with the third power of the size, i.e. $\gamma \sim r^3$. Therefore the dissipated mechanical power will scale as $T\omega = T^2/\gamma \sim \beta(r)^2/r$. In order to evaluate $\beta(r)$, we first note that the total transmission over a semicircular arc can be written as $\exp[-\alpha(R)\pi R]$ with $\alpha(R)$ the loss coefficient per unit length (see inset in Fig. 2b). We then obtain the fraction of transmitted power over the curved part of each arm in our structure as $\beta(r) = \exp[-\alpha(R)L]$ with arm length $L = (5/4)\pi R$ and $R = r/(1 + \sqrt{2})$. We can then plot the expected mechanical power $\sim \beta(r)^2/r$ as a function of rotor size $r$ as shown in Fig. 3a. We select the value $r = 24$ μm which is close to the maximum and corresponds to $R = 10$ μm. With this choice we expect to have an attenuation

coefficient $\alpha(R) = 0.0176\,\mu m^{-1}$ (see Fig. 2b) and therefore we have $\beta(r) = 0.5$. Figure 1c shows a 3D model of the full structure. In addition to the rotor (in green), we designed a static support (in blue) that prevents structure from falling and, once actuated with light, counteracts the recoiling force arising from the absorption of linear momentum in the axial direction. The structures were fabricated by the two photon polymerization of SU8 photoresist in a custom built setup[21]. Resulting 3D structures were imaged by scanning electron microscopy as shown in Fig. 1d (more details are provided in the Supplementary Note 1).

**Torque to power performance**. As a preliminary test, we recorded the free Brownian motion of the rotors when immersed in water. The rotor angle $\theta$ was extracted by a feature-tracking algorithm applied to optical microscopy images that were recorded by a digital camera running at 150 frames per second. We used the time tracks $\theta(t)$ to compute the mean square angular displacement that can be fitted by the linear function $2Dt$ to get the angular diffusion coefficient $D$. The rotational drag $\gamma$ is related to the diffusivity $D$ by the Stokes–Einstein relation $\gamma = k_B T/D = 151 \pm 15$ pN μm s rad$^{-1}$. When a laser beam (1064 nm) is focused at the input fiber, the structures start to rotate smoothly with a speed that is linearly related to input laser power. Figure 3b shows a bright-field image of a rotating structure (see Supplementary Movie 1). Figure 3c shows the angular displacement $\theta(t)$ of the same rotating structure for different levels of incident laser power. The angle $\theta$ (shown in Fig. 3c) is a linear function of time plus a small periodic modulation. To show this, we compute the time autocorrelation of the normalized angular speed fluctuations $\delta\omega = (\omega - \langle\omega\rangle)/\langle\omega\rangle$ shown in the inset of Fig. 3c. We find that these modulations are power independent and of the order of 10%. Moreover we observe that $\omega$ is higher than the average when the rotor arms are not aligned with the linear polarization direction of the laser. We also fabricated structures with 2 and 3 arms (see Supplemental Information) which are found to be unstable as they often rotate off axis and are much more prone to get stuck on the stator. Extracting angular speeds $\langle\omega\rangle$ and multiplying by the previously determined rotational drag $\gamma$ we obtain the linear torque vs power curve shown in Fig. 3d. Similar results have been obtained for tens of experimental replica of our structures. In particular, for four rotors in the same batch, we found less than 5% deviation in rotational speed for any input power. Fitting data in Fig. 3d with a straight line we obtain a

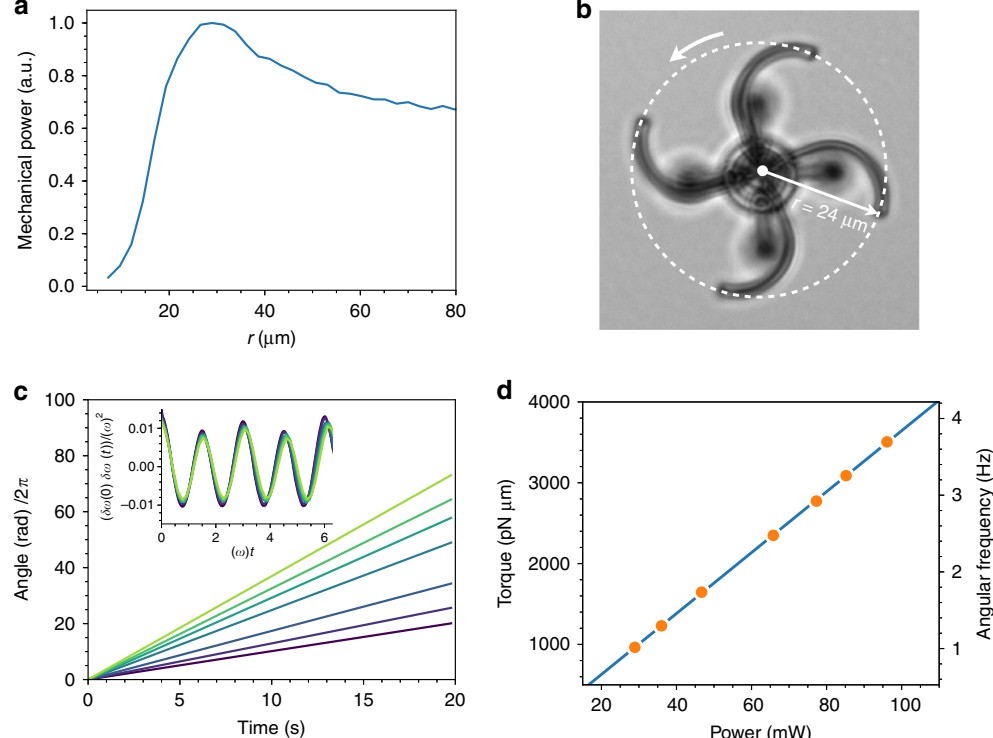

**Fig. 3** Rotating microturbines. **a** Expected mechanical power as a function of the structure radius. **b** Optical microscopy image of a rotating structure. **c** Time evolution of the angular position of a rotating structure driven by different laser powers. Inset shows the time autocorrelation function of the relative fluctuations of the angular speed. **d** Torque and angular frequency as a function of laser power. Laser power has been measured on the objective focal plane with a microscope slide power sensor (Thorlabs S170C). Solid line is a linear fit

torque per unit power coefficient $\tau = T/P = 38 \pm 1$ pN µm mW$^{-1}$. Equation (2) sets an upper bound for this coefficient $\tau_{max} = nr/c = 107$ pN µm mW$^{-1}$ where we have used $n = 1.33$ for water and $r = 24$ µm as the radial distance of the fibers outputs from the structure axis. The ratio $\tau/\tau_{max} = 0.36$ provides a direct experimental value for the torque efficiency in our turbine. The combined effect of coupling, splitting and bend losses then amounts to a total attenuation $\beta(r)\sigma\chi = 0.36$. As discussed above, we can estimate for our structures a value for bend attenuation $\beta(r) = 0.5$. The remaining attenuation can be attributed to the combined effect of coupling and splitting, i.e., $\sigma\chi = 0.72$.

## Discussion

We demonstrated that optical reaction microturbines can exploit guided light propagation to precisely and efficiently reroute an incoming flow of optical energy maximizing the exchanged angular momentum. The resulting torque reaches an unprecedented fraction (36%) of the maximum torque that can be extracted from an optical flow. This limit could be further improved by carefully shaping the driving beam to maximize light coupling with the structure base and minimize splitting losses. Synchronization phenomena in arrays of nearby turbines may also show nontrivial features arising from the interplay of hydrodynamic and optical couplings. Optical micropumps[22], driven by dynamically steered optical traps, have already been employed in microfluidics chips. Optical traps can generate forces of order $QPn/c$ with $Q$ a trapping efficiency factor that is usually in the range $10^{-2}$–$10^{-1}$ (see ref. [23]). In these systems, the theoretical torque efficiency is therefore a fraction $Q$ of $\tau_{max}$ (but in practice[24] often only of order $10^{-6}$) suggesting that optical reaction microturbines could be more efficiently used as flow generators in microfluidic devices.

## Methods

**Fabrication**. For fabrication we use SU-8 2015 photoresist (MicroChem Corp). After exposure the photoresist is baked at 100° for 7 min, then developed by its standard developer solvent and rinsed in a 1:1 mixture of water and ethanol. Strong adhesion of the stator structure to the carrier coverglass is ensured by a layer of the adhesion promoter OmniCoat (MicroChem Corp). After fabrication, the structures are diluted in deionized water.

**Code availability**. The BPM code used in this study is available from the corresponding author on reasonable request.

## Data availability

The data that support the findings of this study are available from the corresponding author on reasonable request.

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

## Acknowledgements

The research leading to these results has received funding from the European Research Council under the European Union's Seventh Framework Programme (FP7/2007–2013)/ERC Grant Agreement No. 307940.

## Author contributions

G.V. and R.D.L. designed experiments. G.V., S.B. and S.F. performed experiments. S.B. performed theoretical analysis of bend losses. C.M. and G.V. analyzed microscopy data. S.B., S.F. and R.D.L. designed simulations. S.B., C.M. and R.D.L wrote the manuscript.

## Additional information

**Competing interests:** The authors declare no competing interests.

