## [Peer Review File · Nature Communications]

Reviewers' comments:

Reviewer #1 (Remarks to the Author):

The paper entitled “An optical reaction micro-turbine” by Silvio Bianchi et al describes construction and performance of a reaction micro-turbine based on in house produced 3D light guiding structure that is able to reroute light the incoming light to multiple output channels. The turbine is made to resemble a “garden sprinkler” and it is stated that this type of a turbine is able to achieve “a strong, uniform and controllable torque and achieve efficiency of 36%.

The paper is written clearly and does convey new and elegant applications of laser written 3D structures to create an efficient turbine. An earlier work by Ormos’ group has demonstrated the garden sprinkler idea on micro scale. Asavei et al also demonstrated a turbine idea on micron scale that was working well and that was quantified. The biggest innovation in the current paper is the elegant coupling of light into photonic fibre-like arms of the sprinkler and guiding the light through them to achieve the effect. A theoretic approach to analysis of the functioning and efficiency of this method is also presented. I think that the paper is elegant and a novel approach to construction of the turbine is presented. The results are elegant and highly convincing. I believe that the theoretical approach is discussing important points. However I slightly question the validity of the conclusions drawn about the bending effects calculation and its application to the current geometry. This is the calculation that gives us the final result about the high efficiency of the turbine (36%). I remain to be convinced about the validity of the approximations made by the authors and I also cannot see the validity of the geometry that is assumed here. – the radial distance of the fibre outputs from the structure outputs and the bend curvature as well as the size of the fibres themselves. Neither can I see how the authors obtain bend attenuation of 0.5. I cannot see any discussion about the other losses that are assumed to be 0.72 and combine the effects of coupling and slitting losses. I think that these should be explain more clearly.

Otherwise I find the paper of high quality and believe that it should be of interest for a broad readership.

Below are just a few more specific comments:

In the paragraph (p2 out of 6)

“Finally, birefringent particles can generate torque through the exchange of the spin component of optical angular momentum [9–11] although the maximum achievable torque in this case is $P\lambda/\pi c$ that is a fraction $\lambda/\pi nr$ of the maximum torque in Eq. 2.”

Authors should add the reference to a paper that first demonstrated transfer of spin angular momentum to birefringent particles, and talked about rotation and alignment that were achieved – M. Friese et al, Nature **394**, pages 348–350 (23 July 1998).

I would also like to draw the authors’ attention to reference 9 in which in fact very high rotation rates have been achieved in contrary to what the authors are stating. Of course, that is more difficult to do in overdamped system. It would be advisable to include a short discussion about this.

In the paragraph (p2 out of 6)

“. We designed a light guiding structure made SU8 ($n_1=1.58$) and immersed in water ($n_0=1.33$) with a chiral shape resembling that of a garden sprinkler. The structure starting with a larger circular core ($4 \mu\text{m}$) oriented in the axial \hat{z} direction (see Fig. 1(c)).”

Should be:

“We designed a light guiding structure made **out of** SU8 ($n_1=1.58$) and immersed in water ($n_0=1.33$) with a chiral shape resembling that of a garden sprinkler. The structure starting with a larger circular core ($4 \mu\text{m}$) oriented in the axial \hat{z} direction (see Fig. 1(c)).”

Reviewer #2 (Remarks to the Author):

The authors describe the development of an optically driven micro-turbine, which is designed as a garden sprinkler to achieve maximal torque. The results are conclusive and the paper is well structured and written. I recommend this work to be published with minor revisions. Here are some further questions and comments:

- Could the authors please extend the introduction to get an overview about potential applications of micro-turbines (e.g. in microrheology or microfluidics)? It is stated that the aim is to achieve maximal optical torque. Why is the power to torque conversion important regarding future applications?
- The authors describe different methods and principles to achieve optical torque and compare the optical torque, e.g., "light mill", "paddle wheel" or the interaction with birefringent particles. However, it is also possible to drive microstructures by using multiple rotating laser spots (see for example: Ito et al. "On-chip fabrication and assembly of rotational microstructures" (2009) [doi:10.1109/IROS.2009.5354506], Maruo et al. "Optically driven micropump produced by three-dimensional two-photon microfabrication" (2006) [doi:10.1063/1.2358820], Maruo et al. "Optically driven viscous micropump using a rotating microdisk" (2007) [doi:10.1063/1.2768631]). What is the estimated optical torque for this method compared to the others? Besides high power to torque conversion, could the authors name other benefits of the presented rotation method regarding applications (e.g. no use of beam modulation)?
- The micro-turbine is fabricated by two-photon polymerization and the SEM image of the structure shows high shape accuracy. How reproducible can functional turbine be made?
- Where is the power measured indicated in Figure 3(d)?
- The rotor is held by the stator, which is fixed to a cover glass. Can the authors estimate, which impact friction or adhesion has on the angular frequency?
- Figure 3(d) shows a linear dependency between laser power and torque. What are the limitations of the system? Will the structure be destructed due to absorption at high laser powers or is the used laser setup the limiting factor?
- The design of the turbine is very elegant und effective. However, three arms would reduce the viscous drag forces and may reduce optical losses, resulting in a higher efficiency. Could the authors specify why four arms were chosen?

Reviewer #3 (Remarks to the Author):

The manuscript by Bianchi et al. describes a novel, light-guiding optical micro-rotor and the analysis of its behavior. A thorough and comprehensive theoretical analysis on light-driven rotation and the functioning of the introduced rotor is presented. The manuscript states that the authors managed to achieve a high (close to maximum) efficiency in driving a rotary motion by light. I find the experimental results convincing, the analyses thorough and accurate, and the conclusions well founded.

In my opinion, the manuscript falls within the interest of the readership of the journal. The topic is interesting also for those outside of the narrow scientific fields involved, the functioning rotor is eye-catching, and the analysis is clever and creative.

I find the manuscript compelling, well structured, comprehensive and very well written. The introduction puts the presented microscopic rotor in context with the general turbine design principles, and compares it with other optical micromotors from the literature. I find that here the paper Galajda, Ormos J. Opt. B-Quantum. S. O. 4 S78-S81 (2002) should be mentioned as a very similar rotor is presented (although light guiding is not discussed in that paper).

The design of the optical turbine is presented at the end of the introduction, and again in greater detail in a separate chapter (Design of an optical turbine). This may seem redundant at first, but I think actually it makes the section in between (Bend losses) more comprehensible. This, in my opinion, approves the structuring of the manuscript.

The parts dealing with theoretical analysis are written in a clear and condensed, but comprehensible manner. The train of thought and logic is comprehensible, and the cited literature provides additional detailed information for a deeper understanding.

The figures are visually appealing and informative, along with the figure captions.

I have a few minor comments only regarding the manuscript.

The parameters P and n in eq. 1. should be explained in the text.

Page 1: ingoing -> incoming

Page 2 Fig.1. captions: computer 3D model -> 3D computer model

Fig.1a-b and Fig. 3b is not referred to in the text.

Page 3: insert "as demonstrated on" or similar in a sentence to read "best match theory and simulation at large R values _as demonstrated on_ Fig. 2(b)."

Page 3 Fig.2. caption: inserting " n " could be useful to read as: "refractive index n profile"

Page 4: insert "the" to read as "_the_ Helmholtz equation"

Page 4: coordinate transform -> coordinate transformation

Page 4: ABCD panel labels are missing on Fig. 3.

Overall I find the manuscript suitable for publication (after addressing the minor comments).

NCOMMS-18-17978

“An optical reaction micro-turbine”

Authors reply

We thank the Reviewers for their quick reply and for appreciating our work. Addressing their comments we have improved the original manuscript by expanding the discussion and by adding experimental data and hydrodynamic numerical simulations (see Supplemental Information). Please find below a detailed, point-by-point reply to the comments raised by the Referees. For clarity, the original Referee comments are included in the reply and typed in *italic*. **Blue text** refers to added/modified sentences as they appear in the revised manuscript.

Reviewer 1 (Remarks to the Author):

1. *The paper entitled “An optical reaction micro-turbine” by Silvio Bianchi et al describes construction and performance of a reaction micro-turbine based on in house produced 3D light guiding structure that is able to reroute light the incoming light to multiple output channels. The turbine is made to resemble a “garden sprinkler” and it is stated that this type of a turbine is able to achieve “a strong, uniform and controllable torque and achieve efficiency of 36%. The paper is written clearly and does convey new and elegant applications of laser written 3D structures to create an efficient turbine. An earlier work by Ormos’ group has demonstrated the garden sprinkler idea on micro scale. Asavei et al also demonstrated a turbine idea on micron scale that was working well and that was quantified. The biggest innovation in the current paper is the elegant coupling of light into photonic fibre-like arms of the sprinkler and guiding the light through them to achieve the effect. A theoretic approach to analysis of the functioning and efficiency of this method is also presented. I think that the paper is elegant and a novel approach to construction of the turbine is presented. The results are elegant and highly convincing. I believe that the theoretical approach is discussing important points. However I slightly question the validity of the conclusions drawn about the bending effects calculation and its application to the current geometry. This is the calculation that gives us the final result about the high efficiency of the turbine (36%). I remain to be convinced about the validity of the approximations made by the authors*

We thank the Reviewer for his/her positive judgment. The Reviewer ex-

pressed concerns about the “validity of the approximation made”. This comment addresses a very important point in our paper and therefore needed to be restated clearly. It should be now clear that the 36% efficiency of the turbine is not the result of a theoretical calculation but a direct experimental result. We now write:

Fitting data in Fig. 3(d) with a straight line we obtain a torque per unit power $\tau = T/P = 38 \pm 1$ pN μ m/mW. Equation 2 sets an upper bound for this coefficient $\tau_{\max} = nr/c = 107$ pN μ m/mW where we have used $n = 1.33$ for water and $r = 24$ μ m as the radial distance of the fibers outputs from the structure axis. **The ratio $\tau/\tau_{\max}=0.36$ provides a direct experimental value for the torque efficiency in our turbines.**

2. *and I also cannot see the validity of the geometry that is assumed here. – the radial distance of the fibre outputs from the structure outputs and the bend curvature as well as the size of the fibres themselves.*

We included as Supplemental Information additional SEM images showing details and measurements of the geometrical parameters of our structures (radial distance of the fibre outputs, bend curvature, fiber thickness).

3. *Neither can I see how the authors obtain bend attenuation of 0.5.*

The bend losses were estimated using the experiments and the simulations results shown in Fig.2(b) we now write more clearly in the text how β has been computed:

We then obtain the fraction of transmitted power over the curved part of each arm in our structure as $\beta(r) = \exp[-\alpha(R)L]$ with arm length $L = 5/4\pi R$ and $R = r/(1 + \sqrt{2})$. We can then plot the expected mechanical power $\sim \beta(r)^2/r$ as a function of rotor size r as shown in Fig. 3(a). We select the value $r = 24$ μ m which is close to the maximum and corresponds to $R = 10$ μ m. With this choice we expect to have an attenuation coefficient $\alpha(R) = 0.0176$ μ m $^{-1}$ (see Fig. 2(b)) and therefore we have $\beta(r) = 0.5$.

4. *I cannot see any discussion about the other losses that are assumed to be 0.72 and combine the effects of coupling and slitting losses. I think that these should be explain more clearly. Otherwise I find the paper of high quality and believe that it should be of interest for a broad readership.*

We think that the Reviewer’s concerns raised above are strictly connected with the misunderstanding that we hope to have now clarified in reply to point 1. The value 36% for the torque efficiency is not a theoretical estimate but a direct measurement. Therefore the value 0.72 is not a theoretical estimate that is used to predict the torque efficiency. The situation is actually reversed: the experimental efficiency, combined with the bend loss value determined as detailed in point 3, is used to evaluate the attenuation due to splitting and coupling losses. This is now better explained as follows:

The combined effect of coupling, splitting and bend losses then amounts to a total attenuation $\beta(r)\sigma\chi = 0.36$. As discussed above, we can estimate for our structures a value for bend attenuation $\beta(r) = 0.5$. The remaining attenuation can be attributed to the combined effect of coupling and splitting, i.e. $\sigma\chi = 0.72$.

5. *Below are just a few more specific comments: In the paragraph (p2 out of 6) “Finally, birefringent particles can generate torque through the exchange of the spin component of optical angular momentum [9–11] although the maximum achievable torque in this case is $P\lambda/\pi c$ that is a fraction $\lambda/\pi nr$ of the maximum torque in Eq. 2.” Authors should add the reference to a paper that first demonstrated transfer of spin angular momentum to birefringent particles, and talked about rotation and alignment that were achieved M. Friese et al, Nature 394, pages 348–350 (23 July 1998).*

We agree with the Reviewer that the paper from M. Friese et al. is an important one in the field and we added it in the references.

6. *I would also like to draw the authors’ attention to reference 9 in which in fact very high rotation rates have been achieved in contrary to what the authors are stating. Of course, that is more difficult to do in overdamped system. It would be advisable to include a short discussion about this.*

It is true that in ref. [3] (ref.9 in the previous version of the manuscript) as well as in ref. [1] the authors observe very high rotations, but this is not in contradiction with the fact that the generated torque is lower than in the cases in which orbital angular momentum is used. Indeed in the above cited papers, high rotational speeds are reached thanks to the low drag of the rotating object. We now remark this fact in the text:

Finally, birefringent particles can generate torque through the exchange of the spin component of optical angular momentum [4, 3, 1, 2]. The maximum achievable torque in this case is $P\lambda/\pi c$ that is only a fraction $\lambda/\pi nr$ of the maximum torque in Eq. (2) (from [2] we compute a torque efficiency of the order of 10^{-3}). Despite a small torque efficiency, high rotation rates can be still obtained with this method provided that the rotational drag is very low as in the case of small particles [3] or low viscosity fluids [1, 2]. Here we are interested in maximizing the efficiency of conversion from optical power to torque so that, in the following, we will only focus on strategies exploiting the orbital angular momentum of light.

7. *In the paragraph (p2 out of 6) “. We designed a light guiding structure made SU8 ($n_1=1.58$) and immersed in water ($n_0=1.33$) with a chiral shape resembling that of a garden sprinkler. The structure starting with a larger circular core ($4\ \mu\text{m}$) oriented in the axial z direction (see Fig. 1(c)).” Should be: “We designed a light guiding structure made out of SU8 ($n_1=1.58$) and immersed in water ($n_0=1.33$) with a chiral shape re-*

sembling that of a garden sprinkler. The structure starting with a larger circular core ($4\ \mu\text{m}$) oriented in the axial z direction (see Fig. 1(c)).”

Text has been changed as suggested.

Reviewer 2 (Remarks to the Author):

1. *The authors describe the development of an optically driven micro-turbine, which is designed as a garden sprinkler to achieve maximal torque. The results are conclusive and the paper is well structured and written. I recommend this work to be published with minor revisions. Here are some further questions and comments: Could the authors please extend the introduction to get an overview about potential applications of micro-turbines (e.g. in microrheology or microfluidics)? It is stated that the aim is to achieve maximal optical torque. Why is the power to torque conversion important regarding future applications?*

We now comment on possible applications to the field of microfluidics in the extended concluding section of the revised manuscript. See reply to the next point.

2. *The authors describe different methods and principles to achieve optical torque and compare the optical torque, e.g., "light mill", "paddle wheel" or the interaction with birefringent particles. However, it is also possible to drive microstructures by using multiple rotating laser spots (see for example: Ito et al. "On-chip fabrication and assembly of rotational microstructures" (2009) [doi:10.1109/IROS.2009.5354506], Maruo et al. "Optically driven micropump produced by three-dimensional two-photon microfabrication" (2006) [doi:10.1063/1.2358820], Maruo et al. "Optically driven viscous micropump using a rotating microdisk" (2007) [doi:10.1063/1.2768631]). What is the estimated optical torque for this method compared to the others? Besides high power to torque conversion, could the authors name other benefits of the presented rotation method regarding applications (e.g. no use of beam modulation)?*

We now discuss rotation through multiple steering beams in the expanded concluding section that also cites the suggested bibliography:

Optical micropumps [7], driven by dynamically steered optical traps, have already been employed in microfluidics chips. Optical traps can generate forces of order QPn/c with Q a trapping efficiency factor that is usually in the range 10^{-2} – 10^{-1} (see ref. [6]). In these systems the theoretical torque efficiency is therefore a fraction Q of τ_{max} (but in practice [5] often only of order 10^{-6}) suggesting that optical reaction microturbines could be more efficiently used as flow generators in microfluidic devices.

3. *The micro-turbine is fabricated by two-photon polymerization and the SEM image of the structure shows high shape accuracy. How reproducible can*

functional turbine be made?

In the text we added a few lines where we comment the reproducibility of our system:

Similar results have been obtained for tens of experimental replica of our structures. In particular, for four rotors in the same batch, we found less than 5% deviation in rotational speed for any input power.

4. *Where is the power measured indicated in Figure 3(d)?*

Laser power has been measured after it has been transmitted by the objective. We used a Thorlabs S170C detector, consisting of a photodiode protected by a coverslip, that is specifically designed for measuring optical power in the focal plane of microscope objectives. We added this information in the caption of Fig.3

Laser power has been measured on the objective focal plane with a microscope slide power sensor (Thorlabs S170C).

5. *The rotor is held by the stator, which is fixed to a cover glass. Can the authors estimate, which impact friction or adhesion has on the angular frequency?*

To evaluate the impact of the friction due to the stator we report in the Supplemental Information additional experiments and simulations. In the SI we show rotors with two and three arms and, with the support of a Rotne-Prager simulation, we demonstrated that the impact on the drag due to the proximity of the rotor to the stator is negligible.

6. *Figure 3(d) shows a linear dependency between laser power and torque. What are the limitations of the system? Will the structure be destructed due to absorption at high laser powers or is the used laser setup the limiting factor?*

In the range of powers we could explore we saw no heating effect due to the very low absorption of SU-8 at the used laser wavelength (1064 nm). However we agree with the Referee that at some very high power heating effects will be the main limiting factor of our light-driven rotors. Before structures are destroyed/melted we expect to see a significant generation of gas bubbles due to the temperature increase, these would cause a very irregular rotation of our structures.

7. *The design of the turbine is very elegant und effective. However, three arms would reduce the viscous drag forces and may reduce optical losses, resulting in a higher efficiency. Could the authors specify why four arms were chosen?*

As we now say in the revised manuscript:

We also fabricated structures with 2 and 3 arms (see Supplemental Information) which are found to be unstable as they often rotate off axis and are much more prone to get stuck on the stator.

Reviewer 3 (Remarks to the Author):

1. *The manuscript by Bianchi et al. describes a novel, light-guiding optical micro-rotor and the analysis of its behavior. A thorough and comprehensive theoretical analysis on light-driven rotation and the functioning of the introduced rotor is presented. The manuscript states that the authors managed to achieve a high (close to maximum) efficiency in driving a rotary motion by light. I find the experimental results convincing, the analyses thorough and accurate, and the conclusions well founded. In my opinion, the manuscript falls within the interest of the readership of the journal. The topic is interesting also for those outside of the narrow scientific fields involved, the functioning rotor is eye-catching, and the analysis is clever and creative. I find the manuscript compelling, well structured, comprehensive and very well written. The introduction puts the presented microscopic rotor in context with the general turbine design principles, and compares it with other optical micromotors from the literature. I find that here the paper Galajda, Ormos J. Opt. B-Quantum. S. O. 4 S78-S81 (2002) should be mentioned as a very similar rotor is presented (although light guiding is not discussed in that paper). The design of the optical turbine is presented at the end of the introduction, and again in greater detail in a separate chapter (Design of an optical turbine). This may seem redundant at first, but I think actually it makes the section in between (Bend losses) more comprehensible. This, in my opinion, approves the structuring of the manuscript. The parts dealing with theoretical analysis are written in a clear and condensed, but comprehensible manner. The train of thought and logic is comprehensible, and the cited literature provides additional detailed information for a deeper understanding. The figures are visually appealing and informative, along with the figure captions.*

We thank the Reviewer for suggesting us the reference Galajda, Ormos J. Opt. B-Quantum. S. O. 4 S78-S81 (2002). We listed it among the light mills references. Despite a sprinkler shape, these structures do not work as reaction turbines. As also stated by the authors, the structure's working principle is based on scattering (as in impulse turbines) since guiding is strongly suppressed by an abrupt 90 degrees bend connecting the central axis to the arms. As a confirmation of that, from data reported in this work we estimated a torque efficiency below 1.5% which falls in the range of efficiencies of other lightmills.

2. *I have a few minor comments only regarding the manuscript.*
 - a) *The parameters P and n in eq. 1. should be explained in the text.*
 - b) *Page 1: ingoing \rightarrow incoming*
 - c) *Page 2 Fig.1. captions: computer 3D model \rightarrow 3D computer model*
 - d) *Fig.1a-b and Fig. 3b is not referred to in the text.*
 - e) *Page 3: insert "as demonstrated on" or similar in a sentence to read "best match theory and simulation at large R values as demonstrated on*

Fig. 2(b).”

f) Page 3 Fig. 2. caption: inserting “(n)” could be useful to read as: “refractive index (n) profile”

g) Page 4: insert “the” to read as “the Helmholtz equation”

h) Page 4: coordinate transform \rightarrow coordinate transformation

i) Page 4: ABCD panel labels are missing on Fig. 3.

Overall I find the manuscript suitable for publication (after addressing the minor comments).

We thank the Reviewer for having carefully read our manuscript and pointed out many small corrections that have been now fixed.

References

- [1] Yoshihiko Arita, Michael Mazilu, and Kishan Dholakia. Laser-induced rotation and cooling of a trapped microgyroscope in vacuum. *Nature communications*, 4:2374, 2013.
- [2] Yoshihiko Arita, Andrew W McKinley, Michael Mazilu, Halina Rubinsztein-Dunlop, and Kishan Dholakia. Picoliter rheology of gaseous media using a rotating optically trapped birefringent microparticle. *Analytical chemistry*, 83(23):8855–8858, 2011.
- [3] Alexis I Bishop, Timo A Nieminen, Norman R Heckenberg, and Halina Rubinsztein-Dunlop. Optical microrheology using rotating laser-trapped particles. *Physical Review Letters*, 92(19):198104, 2004.
- [4] MEJ Friese, TA Nieminen, NR Heckenberg, and H Rubinsztein-Dunlop. Optical alignment and spinning of laser-trapped microscopic particles. *Nature*, 394(6691):348, 1998.
- [5] Masaki Ito, Masahiro Nakajima, Hisataka Maruyama, and Toshio Fukuda. On-chip fabrication and assembly of rotational microstructures. In *Intelligent Robots and Systems, 2009. IROS 2009. IEEE/RSJ International Conference on*, pages 1849–1854. IEEE, 2009.
- [6] N Malagnino, G Pesce, A Sasso, and Ennio Arimondo. Measurements of trapping efficiency and stiffness in optical tweezers. *Optics Communications*, 214(1-6):15–24, 2002.
- [7] Shoji Maruo and Hiroyuki Inoue. Optically driven micropump produced by three-dimensional two-photon microfabrication. *Applied Physics Letters*, 89(14):144101, 2006.

REVIEWERS' COMMENTS:

Reviewer #1 (Remarks to the Author):

the revised manuscript adequately addresses all referees' comments and introduces changes that were suggested. The paper is now of publishable quality. I recommend publication.

Reviewer #2 (Remarks to the Author):

The revised manuscript well responds to all reviewer comments. Inconsistencies have been deleted and the references to existing literature is more comprehensive in the current version. I do not see any further potential for improvement, I recommend this original and well presented work for publication in Nature Comm. It also fulfils the high quality demands of the journal, numerous citations of this manuscript might occur in future.

Reviewer #3 (Remarks to the Author):

I accept the replies of the authors, and I'm satisfied with the changes made in the manuscript. I support the publication of the manuscript in its current state.